# Pharmacological Treatment of Acute Psychiatric Symptoms in COVID-19 Patients: A Systematic Review and a Case Series

**DOI:** 10.3390/ijerph19094978

**Published:** 2022-04-20

**Authors:** Claudia Carmassi, Bruno Pacciardi, Davide Gravina, Sara Fantasia, Gennaro De Pascale, Salvatore Lucio Cutuli, Carlo Antonio Bertelloni, Liliana Dell’Osso

**Affiliations:** 1Psychiatric Unit, Department of Clinical and Experimental Medicine, AOUP, University of Pisa, 56126 Pisa, Italy; claudia.carmassi@unipi.it (C.C.); pacciardib@libero.it (B.P.); dr.fantasiasara@gmail.com (S.F.); carlo.ab@hotmail.it (C.A.B.); liliana.dellosso@unipi.it (L.D.); 2Department of Emergency, Catholic University of the Sacred Heart, 00168 Rome, Italy; gennaro.depascale@unicatt.it (G.D.P.); sl.cutuli@gmail.com (S.L.C.)

**Keywords:** psychomotor agitation, delirium, acute psychosis, restlessness, COVID-19 (coronavirus disease 2019), SARS-CoV-2

## Abstract

Delirium and psychomotor agitation are relevant clinical conditions that may develop during COVID-19 infection, especially in intensive care unit (ICU) settings. The psychopharmacological management of these conditions is receiving increasing interest in psychiatry, considering hyperkinetic delirium as one of the most common neuropsychiatries acute consequences in COVID-19 recovery patients. However, there are no actual internationally validated guidelines about this topic, due to the relatively newly introduced clinical condition; in addition, a standardized psychopharmacologic treatment of these cases is a complex goal to achieve due to the risk of both drug–drug interactions and the vulnerable conditions of those patients. The aim of this systematic review and case series is to evaluate and gather the scientific evidence on pharmacologic handling during delirium in COVID-19 patients to provide practical recommendations on the optimal management of psychotropic medication in these kinds of patients. The electronic databases PubMed, Embase and Web of Science were reviewed to identify studies, in accordance with the PRISMA guidelines. At the end of the selection process, a total of 21 studies (*n* = 2063) were included. We also collected a case series of acute psychomotor agitation in COVID-19 patients hospitalized in ICU. Our results showed how the symptom-based choice of the psychotropic medication is crucial, and even most of the psychotropic drug classes showed good safety, one must not underestimate the possible drug interactions and also the possible decrease in vital functions which need to be strictly monitored especially during treatment with some kinds of molecules. We believe that the evidence-based recommendations highlighted in the present research will enhance the current knowledge and could provide better management of these patients.

## 1. Introduction

Coronavirus disease 2019 (COVID-19) is a pandemic infection caused by a novel strain of coronavirus named Severe Acute Respiratory Syndrome Coronavirus 2 (SARS-CoV-2). Coronaviruses are single-stranded RNA viruses and several subtypes affecting humans have been identified, most of which cause upper respiratory tract infections in immunocompetent individuals [1]. Already known strains of coronavirus caused the severe acute respiratory syndrome (SARS) outbreak, starting in 2002, and the Middle East’s respiratory syndrome (MERS) outbreak, starting in 2012 [2]. The outbreak of SARS-CoV-2 was declared a pandemic by WHO in March 2020. Although most prominently associated with pulmonary manifestations, COVID-19 is increasingly implicated in neuropsychiatric complications, including delirium and psychosis [3]. This seems to be due to the neurotropic properties of the virus [4] leading to a direct infiltration damage during the central nervous system (CNS) invasion and an indirect damage due to the cytokine spread caused by the dysregulation of the inflammatory responses who could exacerbate the neurocognitive impairment [5,6,7].

Given the rapid diffusion of the COVID-19 virus and the increase in social isolation, it has become necessary to implement the at-distance consultation and tele-psychiatry assessment for mild clinical cases [8,9] and, on the other hand, to take care of acute psychopathological symptoms in patients admitted to hospitals where the COVID-19 infection is managed and treated in specific wards [10].

Delirium and psychomotor agitation are significant clinical conditions that may develop during COVID-19 infection, especially in intensive care unit (ICU) settings, in patients with acute respiratory distress and in isolation environments [11]. Delirium is defined as a state of acute confusion presenting with a change in mental status, associated with an altered level of consciousness, impaired attention and concentration, and disorganized thinking or perceptual disturbance until hallucinations, illusions, and misinterpretation of senses [12,13]. 

Episodes of psychomotor agitation or hyperkinetic delirium during COVID-19 infections are among the most frequent psychiatric conditions, with an incidence of approximately 65–80% in the ICU [14], and in these cases the patient may be particularly at risk if acute psychiatric symptoms are not immediately treated, and their medical treatment promptly restored.

Psychiatric assessment and psychopharmacological treatment of patients hospitalized for complications of COVID-19 infection may be necessary when acute psychiatric conditions interfere or, in some cases, make impossible to move forward with medical assistance.

Most severe cases of COVID-19-related pneumonia are treated in anesthesiologic and reanimation wards or in the ICU, where acute psychopathological symptoms may complicate the course of the infective disorder; in fact, delirium is associated with prolonged hospitalization, long-term cognitive and functional impairment, and increased mortality [14,15,16].

The goal of the psycho-pharmacological treatment of the acute psychopathological symptomatology is a first stabilization on a psychic and behavioral level, that will allow the prosecution of the normal therapeutic process. 

The general medical condition of COVID-19-infected patients with symptomatic pneumonia and the necessary medical treatment, do not allow psychiatrists to use all available psychotropic drugs to treat acute psychiatric symptoms. Severe impairment of respiratory function, QTc prolongation, and drug–drug interactions are just some of the many issues that clinicians must consider when they need to use psychotropic compounds in hospitalized patients with COVID-19 infections.

Given our cursory understanding of the pathophysiology of delirium in patients with COVID-19, treatment decisions must be based on symptom presentation, underlying medical comorbidities, and consideration of medication interactions.

In people with COVID-19, psychotropic medications may interact with the medical treatments, and some of their adverse effects may worsen the course and outcome of the underlying medical condition [17]. It is also worth noticing that available recommendations for drug treatment of delirium in COVID-19 patients are for off-label use and that they have been extrapolated from literature and reflecting the general practice patterns of the different workgroups [18].

All classes of psychotropic medications have potentially relevant safety issues for people with COVID-19. Unavoidably, in clinical practice, the risk of unfavorable outcomes needs to be carefully weighed on a case-by-case basis, considering several co-existing risk factors. 

Moreover, although different safety issues have been explored separately, they are actually broadly overlapping (i.e., respiratory function might be impaired by both the sedative effect of medications and the increased risk for respiratory infections). 

All antipsychotics have warnings or explicit contraindications for the use in people with risk of QTc prolongation and for the association with some of the commonly used anti-COVID medical treatments. From a regulatory standpoint, only a few medications have a marketing authorization for at least one of the conditions considered [19].

Regulatory data indicated that most of the medications considered are off-label in people with COVID-19 and delirium, and their prescription should therefore strictly follow the medico-legal procedures for off-label prescribing, being particularly alert of any unexpected safety issues [20].

This situation applies particularly to people with COVID-19, considering that many medical treatments are similarly being used off-label or compassionately [21].

In its complexity, COVID-19 infection provides a paradigmatic example of how standard treatment procedures, being designed around “average” patients, are hardly applicable as complexity increases.

Given the circumstances, clinicians must be particularly vigilant when initiating psychotropic agents in patients receiving medical drugs for COVID-19 [17]. 

According to Ostuzzi et al., few medications showed potential benefits for the treatment of delirium, and possible benefits only emerged for Quetiapine and Dexmedetomidine in ICU settings [19].

The risk of sedation, and potentially associated respiratory impairment, appears to be higher for first-generation antipsychotics, Benzodiazepines, Dexmedetomidine, and some antidepressants (Mirtazapine and Trazodone). According to some authors, Quetiapine, Risperidone, and Aripiprazole are potentially effective medications for the short-term treatment of hyperactive delirium, and might represent an alternative to conventional treatments, such as Haloperidol in COVID-19 patients [19].

Few studies also suggest that Dexmedetomidine and other second-generation antipsychotics are potentially effective medications for the short-term treatment of hyperactive delirium, and might represent an alternative to conventional treatments, such as Haloperidol [19].

Taking into account anecdotal observations, some patients with COVID-19 delirium appear to have increased rates of myoclonus, rigidity, alogia, and abulia, suggesting a dopamine-depletion state. When there are no absolute contraindications, other authors report the use of low-potency antipsychotics to manage behavioral disturbance [18]. Additionally, in these kinds of low dopamine level-related symptoms, Pimavanserin could be a potential candidate [22].

Nowadays, there is a lack of specific studies on COVID-19-related psychopathology and medical treatments of COVID-19-related psychiatric symptoms are necessarily empirical and based on non-specific evidence. Even less is the available evidence about acute psychopathology and psychotropic use in COVID-19-infected patients hospitalized in ICU.

Therefore, there are no guidelines for the management of delirium in patients with COVID-19, and the evidence base for treatment is exceedingly thin [18].

Based on these premises, to contribute to the assistance of this very specific and only recently defined population of patients, we carried out a systematic review of the scientific literature and we also collected four cases of acute psychopathology in patients hospitalized in an intensive care unit because of COVID-19 infection complications.

## 2. Materials and Methods

### 2.1. Literature Search

This systematic search was conducted in accordance with the Preferred Reporting Items for Reviews and Meta-Analyses Statement (PRISMA) guidelines [23]. We have examined, from 1 September 2021 to 28 March 2022, the following databases: PubMed, Embase, and Web of Science. We developed database-specific search strategies including a combination of controlled vocabulary terms and keywords. The basic search string used towards this review was: (“Psychomotor Agitation” [Mesh] OR “Psychomotor Agitation” [Title/Abstract] OR “Delirium” [Mesh] OR “Delirium” [Title/Abstract] OR “Acute Psychosis” [Title/Abstract]) AND (“COVID-19” [Mesh] OR “COVID-19” [Title/Abstract]). The full search strategies for each of the databases included and the relative database-specific string utilized are provided and viewable in the Appendix A.

### 2.2. Eligibility Criteria

The criteria used to include articles in this review were as follows:Case report or studies including human subjects with a diagnosis of COVID-19 and psychomotor agitation or hyperkinetic delirium;Case report or studies that only included individuals of ages > 17;Articles available in English.

Because we aimed to investigate the therapeutic side effects of psychopharmacological therapy of hyperkinetic delirium in patients with COVID-19, studies or case reports in which the type of psychotropic drug or outcome of the patient was not specified, were excluded.

### 2.3. Study Selection and Data Extraction

The publication selection process and data extraction were independently conducted by two of the authors D.G (Davide Gravina) and S.F. (Sara Fantasia). The reviewers analyzed the title and abstract of the articles excluding duplicates and those that did not meet the inclusion criteria or were not available. Subsequently, full text articles were evaluated and those that did not meet the inclusion criteria were removed. 

The following information was extracted: the name of the first author, year of publication, country of research, and study type. For case reports, we collected patient characteristics (gender, age, preexisting pathological condition), treatment, side effects, and outcome. For the other types of studies, sample size, delirium group size, type of population, previous psychiatric history, mean age, treatment, and results, were extracted. 

Any discrepancy that emerged during the process was discussed and consensus reached.

Any disagreements were discussed and resolved by a third author C.C. (Claudia Carmassi).

### 2.4. Quality Assessment

The quality of the case reports was assessed by a standardized tool adapted from Murad et al. [24]. Furthermore, we used the Quality Assessment Tool for Observational Cohort and Cross-Sectional Studies (QATOCCSS) to assess the quality of the other types of studies [25]. Each study was scored as either “good,” “fair,” or “poor” (see Table 1 and Table 2). The quality assessment was performed by two independent reviewers (D.G. and S.F.) and a third reviewer (C.C.) cross-checked the quality assessment results. Disagreements were discussed and resolved with the research team.

## 3. Results

A total of 2063 results was produced in the primary database research. After that, 1835 articles were removed after titles because they were duplicates (*n* = 1006) or not relevant (*n* = 829), and 172 were removed because of other publication type (*n* = 162) or full text was not available or was not in English (*n* = 10).

Subsequently, 35 publications were excluded because pharmacological therapy was not specified or because of a lack of other eligibility criteria. No suitable articles emerged from manually screening the references cited in the selected publications and in the review. Finally, 21 articles were included in the present review. Assessment for inclusion or exclusion is summarized in a flow diagram according to PRISMA recommendations [23]. The study selection process is outlined in a flow diagram (Figure 1). In total, 21 publications were provided by the search, including 16 case reports, 3 case series, 1 cross-sectional study, and 2 single-center cohort study, ranging from 2020 to 2021. Details of each study included in the review are reported in Table 1 and Table 2.

### 3.1. Case Report and Case Series

#### 3.1.1. Sample

In the present search, the sample is equally divided between male and female (50%) and an average age of 55.96 years was reported; in two articles, the age of the subject was not available.

In 53.57% of the cases, the subjects suffered from a pre-existing pathological condition: in nine cases, psychiatric disorders (one PTSD, four schizoaffective disorder, one history of acute psychotic reaction during febrile state, one paranoid personality disorder, one alcohol use disorder, one schizophrenia), and in eleven cases other systemic diseases (such as hypertension, hypothyroidism, diabetes type 2, coronary artery disease and aortic stenosis, asthma and atopic dermatitis, epilepsy, colon cancer, atrial fibrillation, dementia, congestive heart failure, osteoarthritis, chronic obstructive pulmonary disease, hyperlipidemia, right bundle branch block, pre-eclampsia at30 weeks pregnant). In ten cases, there were comorbidities between the previous disorders and in thirteen cases the subject was previously healthy. 

#### 3.1.2. Psychopharmacological Treatment

In 96.43% of cases, at least one antipsychotic was used, specifically in six cases a single antipsychotic was used, in seven cases a combination of antipsychotics, in eight cases a combination of antipsychotics and BDZ, in one case a combination of antipsychotics with other psychopharmacologic drugs, and in one case a single BDZ. 

The main types of antipsychotics (42.86%) used were Haloperidol (twelve case reports) and other types of antipsychotics were Risperidone (in nine cases), Quetiapine (in seven cases), Olanzapine (in five cases), Clozapine (in two cases), Chlorpromazine (in two cases), Aripiprazole (in two cases), and Ziprasidone (in one case).

Lorazepam was chosen in six of the nine cases in which a BDZ was used; Clonazepam, Midazolam or Diazepam were preferred in the other three. 

#### 3.1.3. Side Effects

In 22 cases from the 28 included in this review, no side effects were reported. In the other six cases, side effects were reported: one pneumothorax, one prolonged QTc, one neuroleptic malignant syndrome, one rash of the right lower extremity, one diffuse rash surrounding injection site, and one case of fever associated with tachycardia, rise in white blood cells, and decline in CK value.

#### 3.1.4. Outcome

A good outcome is reported in 17 cases from the 28 selected, specifically in 5 case a single antipsychotic was used, in 4 cases a combination of antipsychotics, in 6 cases a combination of antipsychotics and BDZ, and in 2 cases a combination of antipsychotics and other psychopharmacologic drugs (Valproic Acid or Gabapentin). Considering the only case about a pregnant patient, the therapy was based on a combination of antipsychotic (Haloperidol) and BDZ (Midazolam) and the patient had remission of acute psychiatric symptoms; however, neonatal death was reported on day 31.

In eight of the remaining ten cases, a worse outcome has been reported: in four of them the psychopharmacological treatment (one with a single BDZ, one with a single antipsychotic, one with a combination of antipsychotics, one with a single antipsychotic plus Trazodone) was insufficient to resolve the delirium. In the other five cases (three cases with a single antipsychotic plus valproic acid and two cases with a combination of antipsychotics), a prolonged hospitalization and the progressive worsening of general conditions were reported. In the last case, the patient was transferred to the ICU and intubated for airway protection and to facilitate sedation; two weeks later, the patient was extubated, and delirium was improved. 

### 3.2. Other Type of Studies

#### 3.2.1. Type

In this review were also included one cross-sectional study and two single-center cohort studies.

#### 3.2.2. Sample

In two studies, the samples included male and female patients admitted because of COVID-19; in the other one, the sample includes older patients admitted because of COVID-19-developing delirium. 

The mean age of all patients of the samples was 61.5.

#### 3.2.3. Psychopharmacological Treatment

In both studies with patients with previous psychiatric history and previous psychopharmacological treatment, a therapeutic change has been required; for the other patients, a new therapy was introduced. In two studies, psychopharmacological therapy was based on antidepressants, BDZ, antipsychotic (mostly Haloperidol), or anticonvulsant; however, in the other studies, Propofol, BDZ (Midazolam or Lorazepam) or Dexmedetomidine were used.

#### 3.2.4. Outcome

The cross-sectional study does not show any statistically significant difference in the length of hospitalization or mortality between patients experiencing delirium and the non-delirium group. In one of the single-center cohort studies, the length of hospitalization, ICU length of stay, and duration of mechanical ventilation were higher in the delirium group; however, in the other single-center cohort study, a mortality rate of 71% has been reported. 

## 4. Case Series

We collected four case reports of acute psychomotor agitation in COVID-19 patients hospitalized in the ICU.

These patients treated with first-generation (low potency) antipsychotics and Trazodone in order to provide further information about the choice of medications for the management of delirium in people with COVID-19.

These patients also received complex medical treatment that included medications used for the COVID-19 treatment (dexamethasone, etc.) as well as anesthetics (Propofol, Remifentanil, Midazolam), first-generation low-potency antipsychotics (Chlorpromazine), and antidepressants (Trazodone) because of a condition of hyperkinetic delirium that made it impossible to proceed with medical treatment.

Psychotropic treatment led to different outcomes improving the medical conditions of the patients; however, in two instances, part of the treatment was not effective and lead to complications.

Based on the reported cases, some clinical considerations about psycho-pharmacological treatment of COVID-19-infected patients are discussed.

### 4.1. Case Scenario 1

This report is about a 59-year-old female patient, who was affected by hypertension and obesity (55.8 kg/m^2^). She was admitted to the ICU for acute respiratory failure due interstitial pneumonia caused by SARS-CoV-2 infection. Accordingly, she received daily dexamethasone 6 mg/dL for 10 days, that caused hyperglycemia and required insulin administration. The patient was intubated and mechanically ventilated for 31 days, during which she was sedated with Propofol (5 mg/min) and Remifentanil (4.95 mcg/min) continuous intravenous infusion. Sedation was titrated to maintain Bispectral Index (BIS) values between 40–60 and Richmond Agitation-Sedation Scale at −5 while paralyzed, in order to prevent undesirable awareness. 

On day 11, Propofol was shifted to Midazolam (15 mg/h) intravenous infusion because of high triglyceride serum concentration (450 mg/dL). The patient was tracheotomized after three weeks of oro-tracheal intubation. Although pulmonary function recovered after 20 days of invasive mechanical ventilation, weaning from the latter was slow and complicated by patient agitation when intravenous sedation was stopped. Confusion assessment method for the ICU (CAM-ICU) revealed hyperkinetic delirium. In light of this view, Chlorpromazine (8.3 mg/h) intravenous infusion was started for five days, which was associated with delirium resolution. The patient was discharged alive from the ICU after 45 days of stay. 

### 4.2. Case Scenario 2

This report is about a 52-year-old male patient, whose past medical history did not reveal comorbidities. He was admitted to the ICU for acute respiratory failure due to interstitial pneumonia caused by SARS-CoV-2 infection. Accordingly, he received daily dexamethasone 6 mg/dL for 10 days. The patient was intubated and mechanically ventilated for 52 days. Moreover, he required ECMO for 27 days due to severe hypoxemia. The patient was sedated with Propofol (3 mg/min) and Remifentanil (4.5 mcg/min) continuous intravenous infusion. Sedation was titrated to maintain Bispectral Index (BIS) values between 40–60 and Richmond Agitation-Sedation Scale at −5 while paralyzed, in order to prevent undesirable awareness. The patient was tracheotomized after three weeks of oro-tracheal intubation. Although pulmonary function recovered after 45 days of invasive mechanical ventilation, weaning from the latter was slow and complicated by patient agitation when intravenous sedation was stopped. Confusion assessment method for the ICU (CAM-ICU) revealed hyperkinetic delirium. In light of this view, Trazodone (50 mg tid for 23 days) and Chlorpromazine (50 mg bid for 5 days) intravenous infusion were started with significant agitation improvement. The patient was discharged alive from the ICU after 56 days of stay. 

### 4.3. Case Scenario 3

This report is about a 54-year-old male patient, whose past medical history did not reveal comorbidities. He was admitted to the ICU for acute respiratory failure due to interstitial pneumonia caused by SARS-CoV-2 infection. The patient was intubated and mechanically ventilated for 20 days, during which he was sedated with Propofol (5 mg/min) and Remifentanil (4.5 mcg/min) continuous intravenous infusion. Sedation was titrated to maintain Bispectral Index (BIS) values between 40–60 and Richmond Agitation-Sedation Scale at −5, while paralyzed, in order to prevent undesirable awareness. Although pulmonary function recovered after 15 days of invasive mechanical ventilation, weaning from the latter was slow and complicated by patient agitation when intravenous sedation was stopped. Confusion assessment method for the ICU (CAM-ICU) revealed hyperkinetic delirium. In light of this view, Trazodone (100 mg bid for 22 days) and Chlorpromazine (6.3 mg/h for 13 days) intravenous infusion were started with significant agitation improvement. The patient was discharged alive from the ICU after 34 days of stay. 

### 4.4. Case Scenario 4

This report is about a 72-year-old male patient, whose past medical history did not reveal comorbidities. He was admitted to the ICU for acute respiratory failure due to interstitial pneumonia caused by SARS-CoV-2 infection. Accordingly, he received daily dexamethasone 6 mg/dL for 10 days. Initially, respiratory failure was managed using noninvasive respiratory supports. However, he developed hyperkinetic delirium (RASS 3, CAM-ICU positive) that was managed with intravenous Midazolam (6 mg) administration. Such intervention ended up in further agitation (RASS 4) in a few minutes, that required profound sedation by Propofol and Remifentanil continuous infusion to target RASS-1. Consequently, he received oro-tracheal intubation and invasive mechanical ventilation to improve severe hypoxemia. Since then, 53 days has passed, and the patient is still admitted to the ICU due to severe clinical conditions.

## 5. Discussion

The aim of this review is to summarize and integrate the existing knowledge on pharmacological treatment of acute psychiatric symptoms in COVID-19 patients. At the actual time of writing, there are no other systematic reviews in literature which collect direct data on these kinds of samples. The challenge is due to both the lack of studies on this topic and the particularly complex management of episodes of hyperkinetic delirium or other forms of acute psychopathology in this peculiar population of patients, given the vulnerability of these subjects to side effects of psychotropic treatments. These premises were confirmed by this search, in fact in just three observational studies we met our inclusion criteria, apart from the case reports. Moreover, of these three studies, one of them showed prolonged length of hospitalization, ICU length of stay, and duration of mechanical ventilation in the delirium group compared to patients without psychiatric acute symptoms [6]. In the second, although it does not have statistically significant results, we found higher mortality in patients experiencing delirium [14]. Finally, in the third one, which included older patients admitted because of COVID-19-developing delirium, a mortality rate of 70% is reported [42].

This vulnerability derives from the poor general medical condition of COVID-19-infected patients (severe impairment of respiratory function, etc.), high chances of comorbidity with other general medical conditions, an often-reduced functionality of the excretory organs and the frequent necessity of medical treatment leading to QTc prolongation and with high probability of drug–drug interactions.

Pharmacologic management should be reserved for patients with severe agitation which would result in the interruption of essential medical therapies (such as mechanical ventilation or dialysis catheters) or result in self-harm, or for patients with extremely distressing psychotic symptoms (such as hallucinations or delusions).

Intramuscular injections of typical antipsychotics and BDZ, given alone or in combination, have been the treatment of choice for psychomotor agitation over the past few decades, with Haloperidol and Lorazepam being among the most widely used agents [43]. The data obtained from this systematic review also confirmed choice in patients affected by COVID-19, although these drugs have questionable tolerability profiles, and their use may be particularly problematic in patients with COVID-19 infections.

In COVID-19 patients receiving pharmacological treatment, a QTc prolongation is possible with several of the drugs commonly used in this disorder (chloroquine/hydroxychloroquine, antibiotics and anti-virals) and QTc prolongation may be worsened by adding psychotropics to the treatment. 

Some of the drugs commonly used to treat COVID-19 infection (such as Lopinavir/Ritonavir, Chloroquine/Hydroxychloroquine, Doxorubicin, Ceftriaxone, and other antibiotics) may have clinically significant interactions with commonly used psychotropics and their interactions with them should be accurately taken into consideration in treatment planning of acute psychiatric conditions.

Particularly, Haloperidol is involved in a potentially lethal cardiac arrhythmia named “torsade de pointes.” This finding led European drug authorities to issue a black box warning. After that, a prolongation of QTc interval in ECG of some individuals treated with Haloperidol was related with fatal torsade de pointes and the use of this drug became strictly regulated, especially in its parenteral formulation [44,45,46].

In COVID-19 patients receiving pharmacological treatment, a QTc prolongation is possible with several of the drugs commonly used in this disorder and QTc prolongation may be worsened by adding psychotropics to the treatment. Among the case reports included in this search, a case of prolongation of QTc interval was reported in a single case after introduction of Haloperidol in psychopharmacological treatment [3].

Therefore, the first tendency would be preferable to avoid Haloperidol and to use, if possible, drugs with a lower risk of QTc prolongation or reserve it to most severe or treatment resistant cases. 

Among typical antipsychotics, Chlorpromazine may be used in this population given its indication including delirium, low propensity for cardiovascular or respiratory complications, and low propensity to pharmacokinetic interactions with drugs commonly used in COVID-19 infection treatment. A significant risk of severe hypotension should be considered and, therefore, QTc and blood pressure should be monitored before and during treatment. As described above, our case series would suggest its possible use in COVID-19 patients, especially when no systemic comorbidities are detectable. During the systematic search, these data were confirmed; in fact, the use of Chlorpromazine in combination with Haloperidol in a previous healthy patient, did not report side effects and had a good outcome [26]. Despite this, another case included in the review showed how its use in combination with Trazodone in a patient with severe systemic diseases could have an insufficient effect on symptom resolution [41].

Benzodiazepines (BDZs) may cause respiratory depression or acute paradox reaction and their use should be limited as much as possible in patients with respiratory complications and even more in COVID-19 patients who are at a high risk of respiratory impairment. When BDZ use is considered clinically necessary, then short-lived molecules should be preferred. Benzodiazepines maintain their role in the treatment of patients with alcohol and substance withdrawal deliriums. This search would be in line with the available data on this topic.

Trazodone, an atypical antidepressant, has been used successfully in the treatment of agitation, even in elderly patients and in cases of delirium where general medical conditions precluded the use of other compounds, as described above. There are extremely interesting preliminary data regarding its efficacy and tolerability in the treatment of behavioral and psychological disorders related to organic pathologies of the nervous system [47,48].

Atypical antipsychotics may also be particularly useful in this population of patients. Literature data begin to report the efficacy and tolerability of atypical antipsychotics in COVID-19 subjects with psychomotor agitation but nowadays there are no specific studies about their use in COVID-19-infected patients. Preliminary evidence supports the use of Quetiapine, Risperidone, and Aripiprazole in COVID-19 patients [19]. These data were also confirmed by the present search, even if efficacy was mainly reported when atypical antipsychotics were used in polytherapy except for Risperidone. In fact, we reported six cases in which antipsychotics were used in monotherapy: a good outcome after treatment with Risperidone was reported in five of them [4,34,35]; otherwise, no resolution of psychiatric acute symptoms was observed with monotherapy using Olanzapine [28], as well as monotherapy with Quetiapine showing an insufficient effect, making it necessary to apply an add-on therapy based on Lorazepam [29].

Valproate should be used as third-line medication in add-on therapy for cases where response to treatment is only partial [32].

Dexmedetomidine is an anesthetic increasingly used for sedation in intensive care patients. Evidence suggests that its use may prevent delirium, but also that postoperative administration of this drug may reduce the incidence of delirium [49,50] and may reduce agitation due to delirium, with better effectiveness and safety than Haloperidol [51]. Even though the evidence of delirium treatment with Dexmedetomidine is limited and needs further investigation, it may be a promising treatment option.

When discussing our results some limitations had to be taken into account. Firstly, during the selection process, we only included articles in the English language, so there is a risk of missing relevant articles. Another potential limitation of our research is that, at the date of the last research study, the literature predominantly included case reports, which are anecdotal and inherently biased.

## 6. Conclusions

Even if evidence-based indications are slowly making headway, there are not current validate guidelines for the management of acute agitation in COVID-19-infected patients.

As delirium is a complex manifestation, key triggers and pathophysiology pathways might significantly differ in different populations, according to many factors (e.g., age, underlying medical and psychiatric conditions, altered states of consciousness). Thus, there are notable limitations in comparing data from such different populations. However, from a pragmatic standpoint, similar medications are generally prescribed to COVID-19 patients with hyperkinetic delirium irrespective of the underlying etiology, probably because they target final common pathways (e.g., dysregulations of dopaminergic, serotonergic, noradrenergic, and GABAergic systems) [52,53].

In this particular and vulnerable group of patients, we suggest an individualized and step-based therapy, considering the symptom presentation and the medical comorbidities of the patient.

Therefore, this systematic review and case reports might generate useful insights for future research on promising interventions. Randomized head-to-head studies enrolling clinical patients are urgently needed to test promising medications with safe profiles for COVID-19-infected patients with delirium.

## Figures and Tables

**Figure 1 ijerph-19-04978-f001:**
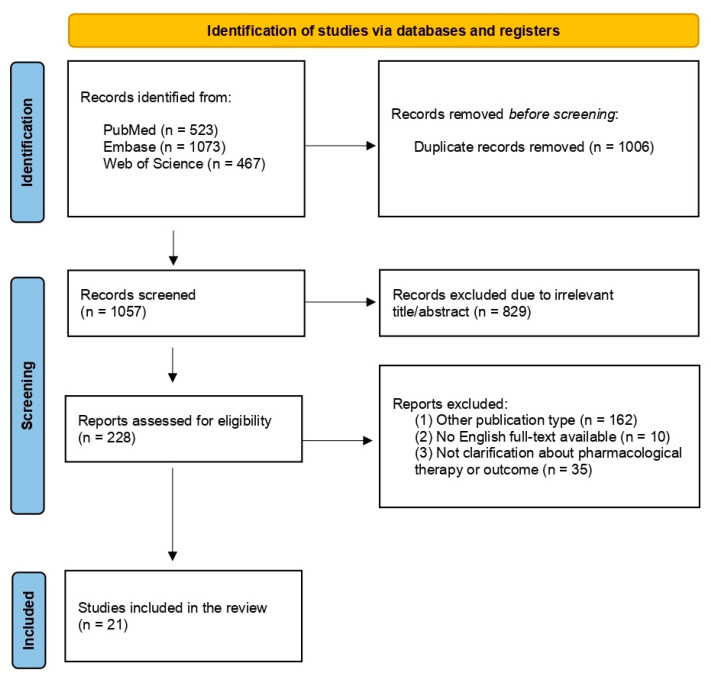
PRISMA flow diagram of the study selection process. PRISMA, Preferred Reporting Items for Systematic reviews and Meta-Analyses.

**Table 1 ijerph-19-04978-t001:** Characteristics of the case report and case series included in the systematic review. PTSD: post-traumatic stress disorder; DM2: diabetes mellitus type 2; HBP: high blood pressure; PPD: paranoid personality disorder; VPA: valproic acid; WBC: white blood cells; CK: creatine kinase; CAD: coronary artery disease; CHF: congestive heart failure; DLB: dementia with Lewy bodies; RBBB: right bundle branch block; COPD: chronic obstructive pulmonary disease.

Study	Date	Country	Type	N° Case	Quality Rating	Previous History	Age	Gender	Treatment	Side Effects	Outcome
Clouden et al. [4]	2020	USA	Case report	1	Good	PTSD	46	F	Risperidone 3 mg	None	Good
Duyan et al. [26]	2021	Turkey	Case report	1	Good	Previously healty	31	F	Haloperidol 10 mg; Chlorpromazine 25 mg	None	Good
Espiridion et al. [27]	2021	USA	Case report	1	Good	Schizoaffective disorder	46	F	Lorazepam 2 mg, Ziprasidone 10 mg; Clozapine 300 mg; Risperidone 8 mg	Neuroleptic Malignant Syndrome	ICU and intubated for 2 weeks
Parker et al. [3]	2021	USA	Case report	1	Good	DM2, HBP	57	M	Haloperidol 5 mg; Lorazepam 2 mg; Aripiprazole 5 mg	QTc Prolungation	Good
Saje et al. [28]	2020	Slovenia	Case report	1	Fair	Acute psychotic reaction during febrile state	Middle-age	M	Olanzapine	None	Insufficient effect
Elfil et al. [29]	2021	USA	Case report	1	Poor	Asthma, atopic dermatitis	20	F	Quetiapine Lorazepam	None	Good
Mahajan et al. [30]	2021	India	Letter to editor/Case report	1	Good	Pre-eclampsia	37	F	Midazolam Haloperidol	None	Agitation remission, neonatal death on day 31
Anmella et al. [31]	2020	Spain	Case report	1	Poor	PPD and DP	68	M	Haloperidol 7.5/24 h IV; Quetiapine 200 mg	None	Good
Sher et al. [32]	2020	USA	Case report	1	Good	Previously healty	70	F	Quetiapine 250 mg; Melatonin 5 mg; VPA IV 1250 mg; Haloperidol IV 8 mg	Pneumo-thorax	Good
Amouri et al. [33]	2020	USA	Case report	1	Good	Previously healty	70	F	Lorazepam IM 0.5 mg	None	Improved catatonia symptoms but no effect on delirium
Mawhinney et al. [34]	2020	UK	Case report	1	Fair	Previously healty	41	M	Olanzapine 10 mg; Clonazepam 2 mg	None	Good
Alonso-Sànchez et al. [35]	2021	Spain	Case series	6	Fair	Previously healty	63	M	Aripiprazole; Quetiapine; Risperidone 6 mg	None	Good
						Previously healty	61	F	Risperidone 6 mg	None	Good
						Previously healty	65	M	Risperidone 6 mg	None	Good
						Previously healty	76	F	Risperidone 6 mg; Gabapentin 900 mg	None	Good
						Previously healty	51	M	Risperidone 4 mg	None	Good
						Previously healty	62	F	Risperidone 3 mg	None	Good
Syed et al. [36]	2021	USA	Case series	4	Good	DM2, schizoaffective disorder, bipolar type	52	F	VPA 1000 mg; Haloperidol IV 2 mg	Tachycardia; Fever; Rise in WBC; Decline in CK value	Prolonged hospitalization and physical deconditioning
						HBP, hyperlipidemia, schizoaffective disorder, bipolar type	61	F	Haloperidol 15 mg; VPA 1000 mg; Benztropine 2 mg	None	Prolonged hospitalization and physical deconditioning
						Colon cancer, DM2, atrial fibrillation, schizoaffective disorder, bipolar type, epilepsy	54	M	Risperidone 2 mg; VPA 500 mg	None	Prolonged hospitalization
						Schizophrenia, hyperlipidemia, hypothyroidism	63	M	Haloperidol 10 mg; Clozapine 350 mg	None	Prolonged hospitalization
Los et al. [37]	2021	Polans	Casereport	1	Good	Previously healthy	39	M	Haloperidol 5 mg IM; Lorazepam 2.5 mg; Olanzapine 10 mg;	None	Good
Gillet et al. [38]	2020	UK	Case report	1	Fair	Previously healthy	37	M	Diazepam; Olanzapina	None	Good
Khatib et al. [39]	2020	Qatar	Case report	1	Good	Epilepsy	52	M	Quetiapine; Haloperidol IM	None	Good
Haddad et al. [40]	2021	Qatar	Case report	1	Good	Previously healthy	Late 30s	F	Lorazepam 2 mg; Quetiapine 300 mg	None	Good
Beach et al. [41]	2020	USA	Case series	3	Good	Dementia, alcohol use disorder, CAD, HBP, atrial fibrillation, CHF	76	M	Olanzapine 2.5 mg; Olanzapine 10 mg; IM, Haloperidol 4 mg IV	Rash of the right lower extremity	Insufficient effect
						DLB, osteoarthritis, HBP	70	M	Trazodone 25 mg; Chlorpromazine 25 mg IV	Diffuse rash surrounding injection site	Insufficient effect
						COPD, DM2, dementia, atrial fibrillation, RBBB, CAD, aortic stenosis, CHF	87	F	Olanzapine 10 mg IM, Haloperidol 1–2.5 mg IV Quetiapine 25–50 mg	None	Improvement in delirium; however, worsening general conditions

**Table 2 ijerph-19-04978-t002:** Characteristics of the other types of studies included in the systematic review.

Study	Date	Country	Type	Quality Rating	Sample	Delirium	Population	Previous Psychiatric History (Delirium Group)	Mean Age (Delirium Group)	Treatment	Results
Arbelo et al. [6]	2020	Spain	Cross-sectional study	Good	71	25	Admitted because of COVID-19	53 (12)	64 (69)	Antidepressant in 8 pts BDZ in 7 pts Antipsychotic in 8 pts Anticonvulsant in 8 pts	Not statistically significant difference
Ragheb et al. [14]	2021	USA	Single-center cohort study	Good	148	108	Admitted because of COVID-19	17 (11)	59 (58)	Propofol, Midazolam, Dexmedetomidine, Lorazepam	↑the median length of stay in delirium group
Rozzini et al. [42]	2020	Italy	Single-center cohort study	Fair	14	14	Older patients admitted because of COVID-19-developing delirium	None	78.2	None in 2 pts Antidepressant in 1 pts BDZ in 4 pts Antipsychotic in 4 ptsUndefined in 3 pts	Mortality rate was 71%

## Data Availability

All data generated or analyzed during this study are included in this published article.

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
