# Peer review of "Pharmacological Treatment of Acute Psychiatric Symptoms in COVID-19 Patients: A Systematic Review and a Case Series"

_ijerph, 2022, doi:10.3390/ijerph19094978_

Round 1
Reviewer 1 Report
The article is actual. And summarizing available knowledge concerning this topic is undoubtedly valuable.
I have two remarks
In Ostuzzi et al review 10 paper were included in authors review 14
The discussion is very similar; I would appreciate to stress what new aspect was brought by this larger a newer review
Further, in the context with the paragraph about dopamine related symptoms (in Introduction) theoretically pimavanserin could be a potential candidate.
Author Response
Point 1: In Ostuzzi et al review 10 paper were included in authors review 14. The discussion is very similar; I would appreciate to stress what new aspect was brought by this larger a newer review.
Response 1: As requested, we stressed in the discussion some relevant newer aspect that differentiate our systematic review to Ostuzzi et al. ones: we collected and analyzed direct data on delirium in patients with a diagnosed COVID-19 disease, and not indirect data of delirium in people with medical conditions or vulnerabilities similar to those of COVID-19 as Ostuzzi et al. did.
Point 2: Further, in the context with the paragraph about dopamine related symptoms (in Introduction) theoretically pimavanserin could be a potential candidate.
Response 2: As suggested, we added that consideration in the introduction paragraph and we also added the relative bibliography reference about that topic.

Reviewer 2 Report
While the aim of the paper is rather interesting, and could help better understand and treat psychomotor symptoms in patients suffering from covid-19, there are several points that should be addressed in order for the paper to be suitable for publication, particularly in the methods section.
Introduction: the content is interesting and the arguments are convincing. But there are too many short paragraphs that reduce the discourse coherence. The paragraphs should be ordered in a more smooth way in order to improve discouse coherence.
Methods:
The first criterion can be included in the second one, just by adding “human” to the latter: Case Report or studies including human subjects with a diagnosis of SARS-CoV- 2 disease and psychomotor agitation or hyperkinetic.
The authors state that the sample included “11 case reports, 1 case series and 2 studies”. The meaning of “studies” is far from clear.
None of the cited works included any control group. Having a control group means that the sample of patients has been randomly distributed into two different independent conditions (2 different treatments, for example). This is not the case in any of the selected papers. Given the recency of covid-19 pandemic, the lack of a control group is not a drawback of these studies. Therefore, I suggest to revise your search criterion, so that the studies not having a control group, but including a description of the treatments administered to the patients and of the outcome, should be now included.
What do the authors mean when stating that “In the 66,67 % of the cases (8 articles), the subject suffered of a pathological condition” Do the authors mean “preexisting pathological condition?
Conclusions
It is not clear what the authors mean with the term “real-world patients”.
General: English editing is required
Author Response
Point 1: The content is interesting and the arguments are convincing. But there are too many short paragraphs that reduce the discourse coherence. The paragraphs should be ordered in a more smooth way in order to improve discouse coherence.
Response 1: As requested, we have tried to re-order some paragraphs and re-writing others to give a more linearity at the discourse.
Point 2: Methods: The first criterion can be included in the second one, just by adding “human” to the latter: Case Report or studies including human subjects with a diagnosis of SARS-CoV- 2 disease and psychomotor agitation or hyperkinetic.
Response 2: We have changed that phrase in light of that consideration, and we added the term “human”.
Point 3: The authors state that the sample included “11 case reports, 1 case series and 2 studies”. The meaning of “studies” is far from clear.
Response 3: We changed the generic term “2 studies” in a more specific and clearer way (“one cross-sectional study and one single-center cohort study”).
Point 4: None of the cited works included any control group. Having a control group means that the sample of patients has been randomly distributed into two different independent conditions (2 different treatments, for example). This is not the case in any of the selected papers. Given the recency of covid-19 pandemic, the lack of a control group is not a drawback of these studies. Therefore, I suggest to revise your search criterion, so that the studies not having a control group, but including a description of the treatments administered to the patients and of the outcome, should be now included.
Response 4: As your suggestion, we revised the search criterion in the materials and methods section and we removed the term “control group”.
Point 5: What do the authors mean when stating that “In the 66,67 % of the cases (8 articles), the subject suffered of a pathological condition” Do the authors mean “preexisting pathological condition?
Response 5: Exactly, that was what we meant; we changed that section adding “preexisting” as your suggestion.
Point 6: Conclusions: It is not clear what the authors mean with the term “real-world patients”.
Response 6: We meant “clinical” patients; we changed that statement to provide a clearer expression form.

Reviewer 3 Report
This manuscript reported a systematic review and case series on pharmacological treatment during delirium in COVID-19 patients. The authors did not follow the Instructions for Authors, especially reference style.
Points to note are:
General
- The authors should report the systemic review following the PRISMA 2020 Statement, not 2009 Statement.
- EMBASE should be Embase
- In general, numbers less than 10 should be written as words.
Abstract
- Lines 12-15: Poor English, please rewrite this sentence.
- Line 18: ‘that patients’ should be ‘those patients’
- Line 30: ‘… to provide an always better care of that patients.’ Poor English, please rewrite this sentence.
Keywords
- ‘SARS-CoV-2 disease’ is incorrect, as SARS-CoV-2 is a virus, not the name of the disease.
Introduction
- Lines 36-37: ‘Severe Acute Respiratory Syndrome 2 Coronavirus (SARS-2-CoV)’ is an incorrect term. It should be SARS-CoV-2
Materials and Methods
- Line 144: PRISMA 2020 Statement should be used.
- Line 145: ‘… from 1 September to 21 October 2021’ is incorrect, as some included studies were published in 2020.
- The search strategy is incorrect, as controlled vocabulary (e.g. MeSH) was not used.
- Full search strategies for all databases should be reported (see PRISMA 2020 Checklist item 7)
- Line 155: ‘SARS-CoV-2 disease’ is incorrect (See Point 7 above)
- Line 161: Covid-19 should be COVID-19 for consistency.
- Figure 1 should use PRISMA 2020 flow diagram, not 2009 version.
- There is no information on the other items within PRISMA 2020 checklist.
Results
- There is no information on the other items within PRISMA 2020 checklist.
Case Series
- Line 268: ‘… Intensive Care Unit (ICU)…’ should be ‘… ICU…’
Discussion
- There is no information on the other items within PRISMA 2020 checklist.
- Has this systematic review been registered? Is there any protocol? (See PRISMA 2020 checklist item 24a, 24b and 24c).
References
- All the references are in the wrong format. There are too many errors to be listed. Only some examples are given below.
- Do not use comma between the authors. Semi-colon should be used instead.
- Do not use ‘&’ before the last author.
- Year of publication should not be placed after the last author and should be in bold.
- Journal name should be abbreviated.
- You should cite all authors, or cite the first ten authors, followed by et al. Therefore, errors in References 8, 9.
- Last page number missing; e.g. References 3, 15, 30, 31, 42.
- Article number missing; e.g. References 4, 9, 24.
- Incorrect page numbers; e.g. References 6, 17, 23.
- Reference 10: Volume number missing.
- Reference 38: ’60-9’ should be ’60-69’
- Reference 43: ‘61’ should be ’61 Suppl 14’
Author Response
Point 1: General
The authors should report the systemic review following the PRISMA 2020 Statement, not 2009 Statement.
EMBASE should be Embase
In general, numbers less than 10 should be written as words.
Response 1: We conducted again the systematic search following the PRISMA 2020 guidelines instead of 2009 Statement (therefore, we have significantly changed part of the materials and methods section, results section, discussion section, references and the diagrams and flow-chart). We also changed the term “EMBASE” in “embase” and re-wtitten all the numbers less than 10 as words as requested.
Point 2: Abstract
Lines 12-15: Poor English, please rewrite this sentence.
Line 18: ‘that patients’ should be ‘those patients’
Line 30: ‘… to provide an always better care of that patients.’ Poor English, please rewrite this sentence.
Response 2: Lines 12-15: we rewrote the sentence: “The psychopharmacological management of these conditions is receiving increasing interest in psychiatry, considering hyperkinetic delirium as one of the most common neuropsychiatries acute consequences in COVID-19 recovery patients.”
Line 18: we changed the sentence in “those patients”
Line 30: we rewrote the sentence: “could provide a better management of that patients”
Point 3: Keywords
‘SARS-CoV-2 disease’ is incorrect, as SARS-CoV-2 is a virus, not the name of the disease.
Response 3: We changed that keyword; the expression has been rewritten as “SARS-CoV-2”.
Point 4: Introduction
Lines 36-37: ‘Severe Acute Respiratory Syndrome 2 Coronavirus (SARS-2-CoV)’ is an incorrect term. It should be SARS-CoV-2
Response 4: As suggested, we rewrote the term in “SARS-CoV-2”.
Point 5: Materials and Methods
Line 144: PRISMA 2020 Statement should be used.
Line 145: ‘… from 1 September to 21 October 2021’ is incorrect, as some included studies were published in 2020.
The search strategy is incorrect, as controlled vocabulary (e.g. MeSH) was not used.
Full search strategies for all databases should be reported (see PRISMA 2020 Checklist item 7)
Line 155: ‘SARS-CoV-2 disease’ is incorrect (See Point 7 above)
Line 161: Covid-19 should be COVID-19 for consistency.
Figure 1 should use PRISMA 2020 flow diagram, not 2009 version.
There is no information on the other items within PRISMA 2020 checklist.
Response 5: Line 144: We changed the literature search using the PRISMA 2020 guidelines.
Line 145: With the statement “from 1 September to 21 October 2021” we meant that the examination and selection process has been conducted during that period; we changed that sentence in a clearer way.
We added that we didn’t use controlled vocabulary as a limitation in the discussion section.
The term “SARS-CoV-2” and “COVID-19” has been changed as suggested.
Figure 1 has been changed in the PRISMA 2020 flow diagram version.
We followed the PRISMA 2020 checklist, and we added the search strategies for databases and information about the items (Study selection, Data Extraction, Quality Assessment, Bias Analysis).
Point 6: Results
Response 6: Line 268: As suggested, we changed the term in “ICU”
Point 7: Discussion
Response 7: We added the information about registration and protocol (PRISMA 2020 checklist item 24a, 24b and 24c): this systematic review hasn’t been registered on PROSPERO.
Point 8: References
All the references are in the wrong format. There are too many errors to be listed. Only some examples are given below.
Do not use comma between the authors. Semi-colon should be used instead.
Do not use ‘&’ before the last author.
Year of publication should not be placed after the last author and should be in bold.
Journal name should be abbreviated.
You should cite all authors, or cite the first ten authors, followed by et al. Therefore, errors in References 8, 9.
Last page number missing; e.g. References 3, 15, 30, 31, 42.
Article number missing; e.g. References 4, 9, 24.
Incorrect page numbers; e.g. References 6, 17, 23.
Reference 10: Volume number missing.
Reference 38: ’60-9’ should be ’60-69’
Reference 43: ‘61’ should be ’61 Suppl 14’
Response 8: all the references have been rewritten following your specifications.

Round 2
Reviewer 2 Report
The authors have addressed the issues stated by the reviewer. In its present form, the manuscript can be useful to guide clinicians decisions
Author Response
Dear reviewer,
thanks for your revision and comments. We are happy that our manuscript’s changes are in line with your suggestions.
Best regards,
Dr. Gravina

Reviewer 3 Report
The authors had attempted to address the comments from peer reviewers. Unfortunately, the methodology is still unsound and therefore the manuscript is not suitable for publication. Points to note are:
Introduction
- Line 143: Hyphen is missing for COVID-19
Materials and Methods
- Response from the authors: “We conducted again the systematic search…” If the authors really conducted the systematic search again following comments from the peer reviewers, why controlled vocabulary was not used in the new systematic search?
- It is not acceptable to cover up the errors in the systematic review by just saying “… controlled vocabulary as a limitation in the discussion section.”
- The search terms must include a combination of keywords and controlled vocabulary (e.g. MeSH). A systematic review without the use of controlled vocabulary must not be published.
- Full search strategies for all databases must be included in the main text or be available in supplementary files in the journal website (PRISMA 2020 checklist item 7). The search strategy of each database must be listed step by step by numbers. A systematic review without full search strategies included must not be published.
Results
- Lines 190-191: The numbers do not add up; 2142 + 1367 = 3509, not 3508.
- Lines 197-198: Do not use the term ‘flowchart’. The term should be ‘flow diagram’
- Line 299: Kg/m2 should be kg/m2
- Line 300: There is no need to use the full term for SARS-CoV-2 as this abbreviation was used before.
References
- Reference 12: ISBN needs to be included [See the Style Guide for MDPI Journals]
- Reference 14: BMJ open should be BMJ Open
- Reference 16: Jama should be JAMA
- Reference 25: Incorrect name of the organisation
Figure 1
- Use flow diagram instead of flowchart
Table 1
- This table is broken at the bottom of page 5.
Author Response
Dear reviewer,
thanks for your comments. We apologize for the delay in the response, but we had to significantly modify some parts of the manuscript to provide a better version in line with the comments. In particular, we conducted again the systematic search with controlled vocabulary terms and keywords for each database (PubMed, EMBASE, Web of Science) and we adopted a database-specific string for each of them. We described the selection process in the “results” section of the manuscript, and we also provided the full search strategies of each database as a supplemental material, as requested.
Following, a point-by-point response to the comments:
Introduction
Point 1: Line 143: Hyphen is missing for COVID-19
Response 1: We added the missing hyphen as suggested.
Materials and Methods
- Response from the authors: “We conducted again the systematic search…” If the authors really conducted the systematic search again following comments from the peer reviewers, why controlled vocabulary was not used in the new systematic search?
- It is not acceptable to cover up the errors in the systematic review by just saying “… controlled vocabulary as a limitation in the discussion section.”
- The search terms must include a combination of keywords and controlled vocabulary (e.g. MeSH). A systematic review without the use of controlled vocabulary must not be published.
- Full search strategies for all databases must be included in the main text or be available in supplementary files in the journal website (PRISMA 2020 checklist item 7). The search strategy of each database must be listed step by step by numbers. A systematic review without full search strategies included must not be published.
Response 2: As said above, we conducted again the systematic search with controlled vocabulary (MeSH) and we added the full search strategies as a supplementary file. We utilized the following controlled vocabulary string for PubMed: (“Psychomotor Agitation” [Mesh] OR “Psychomotor Agitation” [Title/Abstract] OR “Delirium” [Mesh] OR “Delirium” [Title/Abstract] OR “Acute Psychosis” [Title/Abstract]) AND (“COVID-19” [Mesh] OR “COVID-19” [Title/Abstract]) and we had 523 total results. After that, 453 were removed because irrelevant, 18 were removed because other publication types, 5 were removed because was not available the English full text, and 30 were removed because there was a lack of eligibility criteria (pharmacological therapy or outcome were not specified). 17 publications were included in the review. The controlled vocabulary string utilized for EMBASE was: (“Restlessness” OR “Delirium” OR “Acute Psychosis”) AND (“Coronavirus Disease 2019”) and we had a total of 1073 results. After that, 539 were removed because duplicates, 376 were removed because irrelevant, 144 were removed because other publication types, 5 were removed because was not available the English full text, and 5 were removed because there was a lack of eligibility criteria (pharmacological therapy or outcome were not specified). 4 publications more were included in the review. Finally, the database-specific string utilized for Web of Science was: (“Psychomotor Agitation” OR “Delirium” OR “Acute Psychosis”) AND (“COVID-19”). We found a total of 467 records and all of them were removed because duplicates. So, at the end of the selection process we included a total of 21 publications. All the process is described in the manuscript and included in database-specific tables as supplementary files.
Results
- Lines 190-191: The numbers do not add up; 2142 + 1367 = 3509, not 3508.
- Lines 197-198: Do not use the term ‘flowchart’. The term should be ‘flow diagram’
- Line 299: Kg/m2 should be kg/m2
- Line 300: There is no need to use the full term for SARS-CoV-2 as this abbreviation was used before.
Response 3: We changed all the terms as suggested. Obviously, we also changed all the results (percentages and contents) in light of the new literature search and the new articles included.
References
- Reference 12: ISBN needs to be included [See the Style Guide for MDPI Journals]
- Reference 14: BMJ open should be BMJ Open
- Reference 16: Jama should be JAMA
- Reference 25: Incorrect name of the organisation
Response 4: We modified the references as suggested, and we also added in the bibliography the references of the new articles included in the systematic review.
Figure 1
- Use flow diagram instead of flowchart
Table 1
- This table is broken at the bottom of page 5.
Response 5: We modified the term “flowchart” in “flow diagram” as suggested. The table has been significantly modified due to the new selection process. We also provided a not broken table as supplementary materials.
Thank you for the revisions of the manuscript,
Best regards,
Dr. Gravina.
